

# Robust statistical methods and the credibility movement of psychological science

Martina Sladekova and  Andy P. Field

School of Psychology, University of Sussex, Brighton, East Sussex, United Kingdom

## ABSTRACT

The general linear model (GLM) is the most frequently applied family of statistical models in psychology. Within the GLM, the effects under study are estimated using the ordinary least squares (OLS) estimation. In certain situations, OLS produces parameter estimates that are unbiased and optimal (with least possible error) and hypothesis tests that retain the expected rate of false positives (Type I errors). This happens when (1) outliers and influential cases are absent, and (2) assumptions of linearity and additivity, spherical errors, and normal errors are met. This paper first provides a technical description of OLS and an overview of its statistical assumptions. We then discuss the methods commonly employed to detect and address violations of assumptions, and how the current application of these methods can compromise the reproducibility of findings by allowing too many data-driven decisions to be made as part of the data analytic pipeline. We briefly introduce several robust estimation methods—namely bootstrapping, heteroscedasticity-consistent standard errors, $M$-estimators, and trimming—that can improve the accuracy of parameter estimates and the power of statistical tests. We provide guidance on how these methods can be used to transparently preregister a sensitivity analysis, reducing the opportunity for problematic researcher degrees of freedom to enter the analytic pipeline.

Corresponding author
Martina Sladekova,
m.sladekova@sussex.ac.uk

## INTRODUCTION

Even a brief scan of any research literature will reveal an array of commonly-used and seemingly distinct statistical tests such as the $t$-test, analysis of variance (ANOVA), analysis of covariance (ANCOVA), correlation, regression analysis, and a range of multivariate methods.[1] In fact, all of these statistical tests sit within the unifying framework of the general linear model (GLM) (*Cohen, 1968*; *Field, 2024*). Reviews of statistical practice show that this family of models accounts for a major proportion of analyses reported in psychology and psychiatry journals over the years (*Kashy et al., 2009*; *Bakker & Wicherts, 2011*; *Counsell & Harlow Lisa, 2017*; *Nieminen & Kaur, 2019*; also see *Kieffer, Reese & Thompson, 2001* for a historical overview of the use of the GLM in psychology). Within the GLM, the effects

[1]This manuscript was published as a preprint at *Sladekova & Field (2024d)*.

under study are estimated using the ordinary least squares (OLS) method. The goal of OLS estimation is to find an estimate that minimises the sum of squared errors between the observed values and the values predicted by the model. OLS can do so successfully under specific conditions, specifically when (1) outliers and influential cases are absent and (2) assumptions of linearity and additivity, spherical errors, and normal errors are met (*Field, 2024*). We elaborate on what is meant by "successfully" in sections that follow.

Historically, the robustness of OLS models to violations of these assumption has been debated. Early reviews suggested that the *F*-statistic (ANOVA models), the *t*-statistic, and their associated significance tests are unbiased even under non-normality (*Pearson, 1931*; *Lindquist, 1953*; *Boneau, 1960*; *Hsu & Feldt, 1969*) and non-spherical errors (*Box, 1954*; *Scheff'e, 1959*; *Pratt, 1964*). However, more recent work shows that the statistical power of OLS to detect existing effects can be substantially reduced under violations of assumptions, while the parameter estimates can be inaccurate and not representative of a typical individual in a population (*Glass, Peckham & Sanders, 1972*; *Lix & Keselman, 1998*; *Long & Ervin, 2000*; *Hayes & Cai, 2007a*; *Wilcox, 2010*; *Wilcox, 2017*; *Sladekova & Field, 2024a*).

This paper is intended for applied researchers who routinely use OLS estimation in their work and who wish to consolidate their analytic pipelines with recent developments in statistical methodology and open science practices. We first provide a technical revision of OLS models, explaining how some underlying features of OLS estimation relate to its often misunderstood statistical assumptions. We review the impact violations of model assumptions have on parameter estimation and hypothesis testing. We briefly outline the methods commonly employed to detect and address violations of assumptions, and discuss how current statistical practice can compromise the reproducibility of findings by allowing too many data-driven decisions to be made as part of the data analytic pipeline. We introduce several robust statistical methods as alternative and often more suitable estimation procedures that can be used in conjunction with a transparently reported sensitivity analysis to provide more credible account of the findings.

## SEARCH METHODOLOGY

We present a case for the use of robust statistics in situations where researchers typically select OLS models as the default option. This recommendation stems from the review of three related areas answering the questions: (1) Do researchers attend to statistical assumptions? (2) How common are violated assumptions in practice? (3) How do OLS models perform in situations where assumptions are violated?

We searched the databases Scopus, Web of Science and PsycINFO. We applied no limitations on publication date but requested work published in the area of Psychology and relevant sub-disciplines within each database. We only included papers published in the English language. We ran the search for each question separately with a set of search strings and evaluated for inclusion based on separate criteria specified below. During the database search, we manually screened the titles and the abstracts for relevance, followed by an assessment of the full candidate texts. For each paper identified for inclusion *via* database search, we conducted backward search through references cited in a given paper, and

forward search through references citing the paper. We used Google Scholar for forward search, as it provided the most comprehensive list of citing papers, including access to grey literature like preprints and doctoral theses.

The search strings for each research questions were compiled prior to commencing the literature search and not altered during the process other than by selecting relevant filters within each database to apply some of the exclusion criteria (specified for each question below). We kept the search strings purposely broad—this was done to keep the search sensitive and less likely to miss potentially relevant papers while also minimising bias by not prioritising papers reporting certain results over others. The drawback of this strategy was lack of specificity which resulted in broad spectrum of papers exported from each database that required screening. After exporting the list of results from each database, we

1. Manually screened all the titles. The majority of papers excluded at this stage were applied empirical papers that mentioned assumptions checks or parameter estimation methods in the abstract and these papers could be discernibly identified based on the title alone.
2. Assessed the abstracts for papers with potentially relevant or ambiguous titles. Papers were typically excluded on the basis of format (*e.g.*, book chapters) or methodology (see below for exclusions reasons specific to each research question)
3. Assessed any remaining full texts for inclusion based on the criteria below.

(1) **Do researchers attend to statistical assumptions?** We included studies empirically assessing statistical practice relevant to OLS assumptions, either through primary data collection or by evaluating practice as reported in published papers (meta-research). We excluded simulations, methodological reviews, and book chapters. We used the search string *assumption\* AND (model\* OR statistic\* OR linear OR regress\*) AND (check\* OR test\* OR assess\*)*, targeting phrases such as "checking model assumptions", "testing statistical assumptions", or "assessing assumptions of linear models", or alternative permutations of the search terms. We also looked at how researchers handle outliers and influential cases using the string *(outlier\* OR influential AND case\*) AND (psycholog\* OR research\*)*, searching for evidence of either the statistical practice related to outlier handling, or their prevalence in psychological research with respect to research question (2).

(2) **How common are violated assumptions in practice?** We included empirical studies and literature reviews summarising the properties of statistical models used in psychological research, namely sample size, distributional characteristics of variables or model errors, and the presence of outliers or influential cases. If a primary study was subsumed in a literature review, it was not included as additional evidence on its own. We excluded simulations, methodological reviews, and book chapters. We used the following search strings to identify papers summarising typical sample sizes *(sample AND size\*) AND psycholog\**, distributions of variables or model errors *(skew\* OR kurt\* OR normal\* OR non-normal\* OR nonnormal\*) AND (distribut\* OR variable\* OR data) AND NOT (simulat\* OR model\*)* and heterogeneity of variance or heteroscedasticity *(heterogeneity AND of AND variance) OR (heteroscedasticity) OR (variance AND ratio\*)*, targeting phrases like "sample sizes in psychology", "skewed variables", "non-normal distributions", or "heterogeneity of variance".

**(3) How do OLS models perform in situations where assumptions are violated?** We included empirical simulations and literature reviews evaluating OLS performance in conditions where statistical assumptions are violated. Similar to above, if a simulation study was subsumed in a literature review, it was not included as additional evidence on its own. We excluded tutorial papers, methodological papers discussing a method without empirical evaluation, simulations evaluating specific general linear model forms like repeated measures or designs with multiple-dependent variables, as these are not the focus of the present paper, and simulations focusing on alternative estimation methodology like maximum likelihood in generalised linear models. We used the following search string: *(anova OR ols OR regression) AND (robust\* OR assum\* OR \*normal OR heteros\* OR heterog\* OR violat\*)*, allowing us to capture phrases "ANOVA robustness", "regression assumptions", "performance of OLS under violated assumptions" and similar.

The following sections provide a technical description of OLS estimation and its assumptions, followed by a review and a discussion of the research questions presented above.

## HOW MODEL PERFORMANCE IS EVALUATED

A pre-requisite for any successful modelling is ensuring the model is correctly specified. That is, the variables and interactions explaining the outcome are accounted for as the predictors in the model, while irrelevant variables are omitted. Correct model specification should be driven theoretically rather than statistically and is often formulated prior to data collection. This review focuses on aspects of the analysis that can go wrong *after* data collection, and any statements and advice within this paper are therefore made under the principal assumption that the model has been correctly specified to begin with.

When modelling a relationship between variables, we need to choose the most appropriate *estimator*. An estimator, like the OLS estimator or the maximum likelihood (ML) estimator, is a tool used in the estimation process. The exact way in which an estimator works is typically described by an equation—we will explain the inner workings of the OLS estimator in the next section. The resulting value produced by an estimator during the estimation process is called the *estimate*.

Within the frequentist framework, we generally care about three aspects of a model when evaluating whether it is "doing its job"—the estimates of the population parameter, the standard errors, and the *p*-values. The term *bias* is often used with reference to statistical assumptions. That is, if assumptions are violated, the analysis may become "biased". While some aspects of the analysis do indeed become biased, others are affected in different ways that will impact the conclusions we can draw from our models. "Bias" refers to a situation where a value produced by a model systematically deviates from some expected value. A significance test is biased if its observed rate of false positive findings (Type I error rate) differs from a theoretical alpha level. For example, if the theoretical alpha is set at the conventional 0.05, we would expect the significance test to produce a statistically significant result only in 5% of cases under the null hypothesis.

Conversely, an estimator (such as the OLS) is considered biased if it systematically over- or underestimates the population parameter. When we estimate a parameter, our sample

produces a single value from a sampling distribution of many possible values. The estimate from our sample will not always have the exact value as the population parameter, but a sampling distribution produced by an unbiased estimator will be centred on the population value. This will not be the case if the estimator is biased.

What also matters is the width of the sampling distribution—that is, how far from the population parameter do the possible values produced by our estimator spread out. An estimator that produces the narrowest sampling distribution will have the smallest variance, while an estimator with a wide sampling distribution will have a large variance. An estimator with the smallest variance can be considered *optimal*.

Formally, these two qualities can be thought of as the components of the Mean Squared Error (MSE) associated with an parameter, which is often used as a benchmark when evaluating model performance. For example for a parameter $\beta$, the MSE can be decomposed as the sum of the variance and squared bias associated with the parameter:

$$MSE = Var(\beta) + Bias(\beta)^2.$$

OLS will always try to find the smallest possible value for MSE, regardless of whether bias is absent $(Bias(\beta)^2 = 0)$ or present $(Bias(\beta)^2 > 0)$. A non-optimal estimate with a large variance is therefore not necessarily biased—it just means that more samples are needed for the sampling distribution to converge on the population parameter when sampling randomly. Likewise, there are situations where an estimator can be biased, but remain optimal. In statistical terms, when an estimator is both unbiased, and simultaneously yields the smallest variance, it is often said to be BLUE—Best Linear Unbiased Estimator. For the purposes of this paper, we'll use the terms *unbiased* and *optimal* when referring to such scenario.

Finally, the *accuracy* of an estimator can also be called into question. Throughout this paper, we use this term generally to describe a situation where an estimator produces sample estimates that are not a realistic reflection of the processes in the population. A common cause of this is an incorrectly assumed error structure, which can affect various types of estimates. For example, in OLS context, the formula for estimating the standard errors takes certain shortcuts by assuming that the model errors are structured in a certain way. If we take these shortcuts where we shouldn't, the resulting standard error will be inaccurate. What's worse, this will also have knock on effects on confidence intervals and $p$-values which rely on the accuracy of standard errors. Similarly, the parameter estimates may be inaccurate if the estimator expects a symmetrical error distribution while, in reality, the population errors are skewed with asymmetrical tails. In this case, the parameter estimate is still statistically sound, but it may not be practically as useful as an estimate produced by an alternative estimator. We elaborate on these situations with reference to OLS estimation in the sections that follow.

## ORDINARY LEAST SQUARES ESTIMATION

The goal of OLS estimation is to produce parameter estimates that result in the smallest possible sum of squared errors in the model. Under specific conditions, the OLS estimates

will be optimal and the statistical significance tests associated with these estimates will be unbiased. The OLS estimates will also align with maximum likelihood estimates.

The conditions under which bias is minimised and the estimates are optimal include (1) satisfied statistical assumptions, specifically linearity and additivity, spherical errors, normal errors, and normal sampling distribution of the parameter, and (2) absence of outliers and influential cases. Here we briefly describe these conditions and summarises the effects that violated assumptions can have on estimation and inference—for a more detailed account, see *Field (2024)*, *Wilcox (2017)* or *Wilcox (2010)*.

## Linearity and additivity

Broadly speaking, assumptions made by OLS models can be divided into two categories. The first one includes assumptions that OLS shares with other statistical models regardless of the method used to estimate the parameters, like linearity and additivity. The second category concerns assumptions related to the error distribution and structure where, unlike alternative estimation methods, OLS fails to offer any modelling flexibility should violations occur.

OLS models assume that the true, or population, relationship between predictor(s) and the outcome is linear. As an equation, this is expressed as

$$Y_i = \beta_0 + \beta_1 X_{1i} + \varepsilon_i$$
$$\varepsilon_i \sim N(0, \sigma^2)$$

where $Y_i$ is the outcome value for an individual $i$, $\beta_0$ is the unknown value of the outcome when all predictors are 0, and $\beta_1$ is the unknown parameter associated with the predictor $X_{1i}$. This parameter represents the change in the outcome variable associated with a unit change in the predictor (Fig. 1A), which for dummy-coded categorical predictors represents the difference in the mean level of the outcome between one category of the predictor and a reference category (Fig. 1B). The error terms for each $i$ ($\varepsilon_i$), are unobservable but are assumed to be random variables that are normally distributed with a mean of 0 and constant variance of $\sigma^2$. This assumption is made explicit in the second line of the equation. The parameters ($\beta_0$ and $\beta_1$) are also unobservable but can be estimated from data observed in a sample. The resulting values are known as parameter estimates and are typically denoted with hats to remind us that they are estimates based on a sample. The resulting model for a sample is, therefore, expressed as

$$Y_i = \widehat{\beta_0} + \widehat{\beta_1} X_{1i} + e_i$$

in which $e_i$ is the observed residual (the difference between the observed value of the outcome and the value predicted by the model) for entity $i$ (Fig. 2A). When there are several predictors in the model, their combined effect is assumed to be additive, *i.e.*:

$$Y_i = \beta_0 + \beta_1 X_{1i} + \beta_2 X_{2i} + ... + \beta_n X_{ni} + \varepsilon_i$$
$$\varepsilon_i \sim N(0, \sigma^2).$$

The parameter estimate associated with each predictor represents the change in the outcome variable associated with a unit change in the predictor when other predictors are held constant.

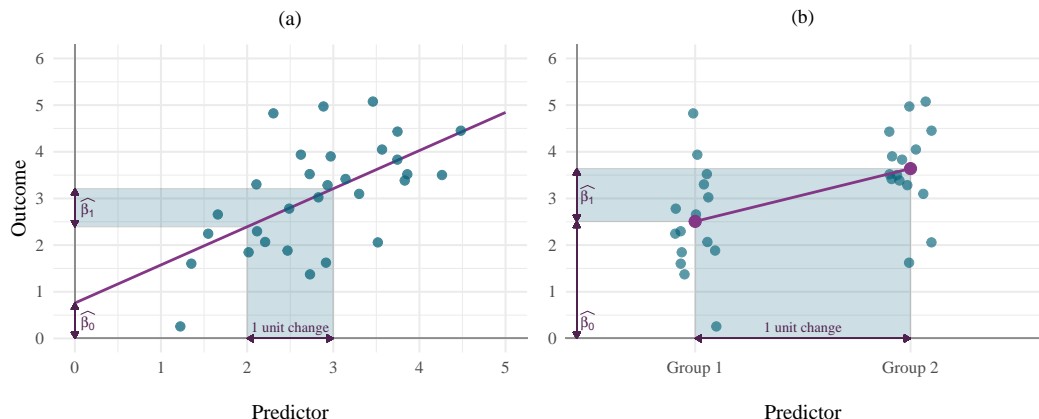

**Figure 1** **Examples of the interpretation of $\widehat{\beta}$s in linear models with (A) continuous and (B) categorical predictors.** For continuous predictors, $\widehat{\beta}_0$ represents the value of the outcome when the predictor is 0, and $\widehat{\beta}_1$ is the change in the outcome associated with one unit change in the predictor. For categorical predictors, $\widehat{\beta}_0$ is the value of the outcome for the baseline category, while $\widehat{\beta}_1$ is the difference between the respective group and the baseline.

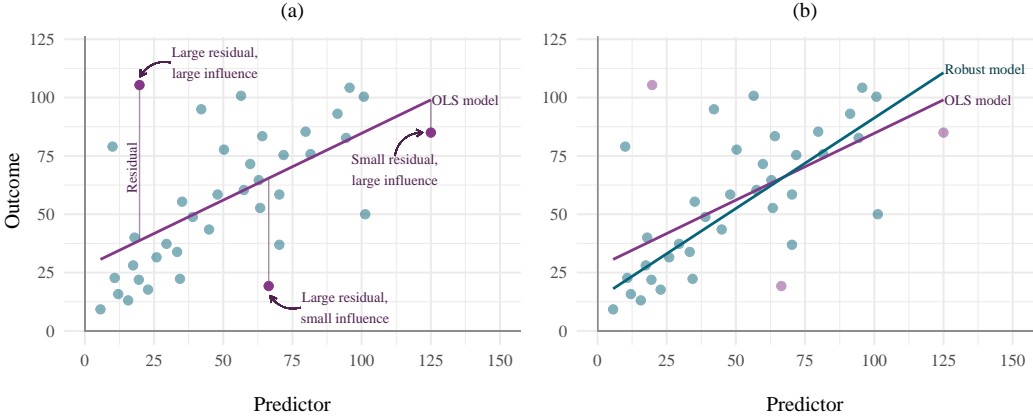

**Figure 2** **Effects of influential cases and outliers on estimation.** (A) Vertical distance of each point from the model line represents the residual. Going from left to right, the highlighted points represent cases with large residual and large influence, large residual and small influence, and small residual and large influence. (B) A comparison of the model fitted using OLS estimation (labelled as 'OLS model') and MM-estimation (labelled as 'Robust model').

Linearity and additivity are crucial prerequisites for modelling linear relationships. This assumption is not unique to OLS models but generalises across estimation methods. Whether researchers apply OLS or opt for alternative methods like Maximum Likelihood Estimation or Bayesian Estimation instead, the primary consideration should be whether the linear and additive form of the model adequately reflects the relationship between the variables of interest. When the relationship being modelled is not, in reality, linear or additive the model is not fit for purpose and any estimation or inference based on it is

invalid (*Gelman & Hill, 2007*). In this sense, linearity and additivity are the most important conditions for a useful model.

## Spherical errors

The Gauss-Markov (GM) theorem states that under certain conditions the OLS estimator for an additive linear model will have lowest sampling variance compared to other unbiased estimators. These conditions are:

1. Errors are, on average, zero. More formally, the expected value of model errors is zero, $E[\varepsilon_i] = 0$.
2. Homoscedastic errors: the variance of errors is a constant, finite, value ($V(\varepsilon_i) = \sigma^2 < \infty$). We have already mentioned that model errors are assumed to be normally distributed *with constant variance* and this condition (referred to as homoscedasticity) reflects the constant variance.
3. Errors are uncorrelated. More formally, $Cov(\varepsilon_i, \varepsilon_j) = 0$, $i \neq j$.

The last two of these conditions are often referred to as the population having 'spherical errors', that is, errors are both independent and homoscedastic (have a constant variance). Non-spherical errors (where either independence or homoscedasticity is not met) do not bias the parameter estimate, however this estimate will not be optimal—it will produce a larger amount of error than if the assumption was met, or if an alternative more robust estimator was applied. Additionally, the standard error will be inaccurate because the formulas for computing standard errors assume independence and constant spread of errors across the levels of the predictors. This in turn affects the confidence intervals and biases the significance tests (*Field, 2024*).

Assumptions of spherical errors and normal errors (see below) are not limited to OLS models, but they can pose a greater logistical challenge for OLS compared to other estimators. Although models using *e.g.*, maximum likelihood estimation or Bayesian estimation often require these conditions to be met, they also allow researchers to select an alternative distribution family for modelling or to define more complex error structures if spherical or normal errors cannot be assumed. For ease of expression, we refer to all the assumptions as "OLS assumptions", however it's worth keeping in mind that other estimators are not immune to their violations.

## Normal errors

Model errors are assumed to be *normally distributed* with constant variance, but as we shall see this assumption has little bearing on estimating model parameters. However, when model errors are normally distributed it can be shown that the parameter estimates have a normal sampling distribution:

$$\widehat{\beta} \sim \mathcal{N}\left(\beta, \sigma^2(\mathbf{X}^\mathrm{T}\mathbf{X})^{-1}\right)$$

in which $\mathbf{X}$ is an $n \times p$ matrix containing a number of rows equal to the number of observations in the sample ($n$), and $p$ columns corresponding to a column of 1s (the intercept) and remaining columns represent values of each predictor variable. Aside from the observation that the sampling distribution is normal when the model errors are normal,

note that the variance of the distribution of $\widehat{\beta}$ is a function of the variance of errors ($\sigma^2$). The variance of errors is unobservable and must be estimated from the observed data ($\sigma^2 = s^2$), giving us the following estimate of the variance of the parameter estimates:

$$\widehat{\mathrm{Var}}\left(\widehat{\beta}\right) = s^2(\mathbf{X}^\mathrm{T}\mathbf{X})^{-1}.$$

The standard error of the parameter estimates is the square root of this estimate

$$\widehat{SE}\left(\widehat{\beta}\right) = \sqrt{s^2(\mathbf{X}^\mathrm{T}\mathbf{X})^{-1}}.$$

It turns out that $s^2$ follows a scaled chi-square distribution, meaning that the standard error of parameter estimates also follows this distribution. Two important statistics related to the parameter estimates depend on their standard error: (1) test statistics; and (2) confidence intervals. To test whether $\beta$ is equal to a particular expected value (typically $\beta_{\mathrm{Expected}} = 0$) we use a test statistic that is the ratio of the difference between the parameter estimate and the expected value to the sampling variation (signal to noise ratio):

$$T = \frac{\widehat{\beta} - \widehat{\beta}_{\mathrm{Expected}}}{\widehat{SE}\left(\widehat{\beta}\right)}.$$

As noted earlier, when errors are normally distributed parameter estimates have a normal sampling distribution. Because of the central limit theorem parameter estimates also have a normal sampling distribution when sample sizes are large (more formally, as $n \to \infty$). In these two scenarios, the test statistic above is the result of dividing a normally distributed variable by one that has a scaled chi-square distribution (see above). Dividing a normal distribution by a scaled chi-square distribution results in a $t$-distribution when the null hypothesis is true, and a non-central $t$-distribution when it is not. Therefore, under the aforementioned conditions the test-statistic follows some form of $t$-distribution. This fact is used to construct confidence intervals around the parameter estimate

$$\widehat{\beta}_{\mathrm{CI\ boundary}} = \widehat{\beta} \pm t_{1-(\alpha/2)} \times \widehat{SE}\left(\widehat{\beta}\right).$$

where the critical $t$ value for given alpha level is multiplied by the standard error estimate of of the parameter estimate $\widehat{\beta}$ and then subtracted or added to $\widehat{\beta}$ to create the interval.

To summarise, the assumption of "normality" refers to the requirement of OLS for a normal sampling distribution of $\widehat{\beta}$. If the errors in the population model are normally distributed, this requirement will be met. If the errors are not normally distributed, normal sampling distribution of $\widehat{\beta}$ can only be assumed in samples that are sufficiently large to invoke the Central Limit Theorem. If the parameter estimate does not come from a normal sampling distribution, the standard errors cannot be estimated accurately and the hypothesis tests will be biased. Additionally, if the population errors are not normally distributed, the OLS estimator is not optimal. Note that under the violations of both the spherical errors and normality, the parameter estimates remain unbiased.

## Absence of outliers and influential cases

Outliers are typically defined as extreme cases with a large residual (*Darlington & Hayes, 2017*). The term 'residual' is equivalent to model *error* when we're talking about a sample

model rather than a population model. Simply put, it is the distance between the outcome value predicted by the model and the actual outcome value observed in a sample. Because the goal of the OLS estimator is to always minimise the amount of error in the model, the estimates produced by a model containing outliers will still be optimal, however they will be biased in the direction of the outlier. Influential cases are cases that have substantial influence on the parameter estimates. Influential cases can often be less obvious than outliers, especially if they have enough influence to bias the estimate to the point where their residual is minimised (Fig. 2A) and would not be detected as an outlier using conventional methods.

## OVERVIEW OF CURRENT PRACTICE

OLS assumptions are often misunderstood and overlooked in statistical analyses. Studies looking at reporting practices show that vast majority of papers don't report any assumption checks (*Counsell & Harlow Lisa, 2017*). Reports of outlier detecting practices are also rare, where only between 7–9% show evidence of outlier handling (*Bakker & Wicherts, 2014*; *Zanin, Lóczi & Zanin, 2024*), although both *Matthes et al. (2015)* and *Valentine et al. (2021)* note an increase in the likelihood of reporting overtime. In the rare instances where assumptions are reported, there's evidence of misconceptions about them (*Ernst & Albers, 2017*).

Granted, lack of reporting is not necessarily indicative of insufficient attendance to the problem, given that the decision to include assumption checks in the final manuscript write-up can be down to external factors, like word count limitations or editorial guidelines. Additional evidence comes from primary studies that either ask researchers about their practice directly or presents them with an analysis exercise to see whether assumption checks are carried out as part of the analytic pipeline. The most optimistic picture is painted by *Dickwelle Vidanage (2022)* who showed that about half of the participants mentioned assumption checks when analysing data, however majority of them failed to detect violations in an analysis exercise. In a sample of PhD researchers, *Hoekstra, Kiers & Johnson (2012)* found that only about 1/3rd of participants attend to assumptions; similar findings have been reported for a broader sample including post-doctoral researchers and members of the faculty (*Sladekova, Poupa & Field, 2024*). Self-reports of researchers corroborate these findings, while also highlighting some fundamental misunderstandings about the nature of the assumptions (*Sladekova & Field, 2024b*). Outliers tend to be attended to more often compared to statistical assumptions—in *Sladekova & Field (2024b)*, we found that up to 50% of researchers report checking for the presence of outliers, while only about ~30% attend to normality or homoscedasticity. However, 2/3rds of researchers neglect diagnosing influential cases which would not necessarily be dected using conventional outlier-detection methods based on how extreme a case is compared to the rest of the data (*Sladekova & Field, 2024b*).

The minority of researchers who do attend to the assumptions either focus on incorrect parts of the model—for example they will check the distributions of the predictor or the outcome variables, even though the assumptions refer to the population model errors—or they tend to use methods that perform poorly in applied settings (*Sladekova & Field, 2024b*;

*Sladekova, Poupa & Field, 2024*). For example, significance tests—like the Shapiro–Wilk test (*Shapiro & Wilk, 1965*), the Kolmogorov–Smirnov test (*Massey, 1951*; *Lilliefors, 1967*) or the Levene's test (*Levene, 1960*)—are a popular method for detecting the presence of non-normality or heterogeneity of variance. In this framework, a statistically significant assumption test is indicative of a violation. A common strategy in this scenario is to then apply a non-paremetric method to test the hypothesis, like the Wilcoxon-Mann-Whitney test (*Mann & Whitney, 1947*) or the Kruskal-Wallis test (*Kruskal & Wallis, 1952*). This strategy comes with a handful of pitfalls. Like any hypothesis tests that rely on *p*-values, assumption tests lack power in small samples and are overly sensitive in large sample where violated assumptions are less of a concern (*Wilcox, 1996*; *Field, 2024*). Given large enough sample, a frequentist hypothesis test will always be statistically significant as no effect is truly zero and no distribution is perfectly normal or homoscedastic (*Rochon, Gondan & Kieser, 2012*). What's more, using null hypothesis significance testing in this way creates the inverse probability fallacy—a non-significant *p*-value *cannot* provide evidence for the absence of an effect, yet it is routinely used to indicate absence of assumption violations. If the decision making about the absence or presence of an effect is conditional on the result of an assumption test, the rate of false-positive findings becomes inflated in the long run (*Gans, 1981*; *Long & Ervin, 2000*; *Zimmerman, 2004*; *Hayes & Cai, 2007b*; *Ng & Wilcox, 2011*). Further, transforming scores into ranks (as is the case for classic non-parametric tests) changes the hypothesis being tested into one that no longer maps onto the data that were originally collected, and distorts the ability of the analyst to meaningfully interpret the results. Similar challenge arises if variables are transformed prior to being included in an OLS model, for example using log-transformation (*Osborne, 2002*; *Grayson, 2004*; *Schmidt & Finan, 2018*). Finally, classic rank-based tests are only available for a limited range of designs, often forcing researchers to either simplify their factorial designs into one-way models, or ignore violations and rely on a biased hypothesis test.

A potential way of overcoming difficulties with violated assumptions while retaining the interpretability of parameter estimates is to use Bayesian estimation. As previously highlighted, models that use Bayesian estimation are still affected by non-normal and non-spherical errors, however this requirement can be relaxed by selecting an alternative distributional family and error structure for modelling the relationship of interest. For example, researchers could select distributions from the exponential family to model skewed outcomes—similar benefit can be obtained by applying maximum likelihood estimation in place of OLS. In addition, Bayesian models provide a probabilistic representation of the parameter estimates drawn from posterior distributions. This means that the parameter estimate itself represents the most likely value (given the data), while the intervals constructed around the estimate tell us where the population parameter is likely to fall with a specified probability allowing for a more intuitive way of handling uncertainty (*Morey et al., 2016*). This is in stark contrast with frequentist confidence intervals which are not only directly affected by violated assumptions, but are also often subject to misinterpretation by the researchers who use them (*Hoekstra et al., 2014*). The drawback is that the use of Bayesian models does not merely entail switching to a different estimation method, but also requires a philosophical shift in approach to hypothesis testing which comes with its

own set of challenges (*Lakens, 2021*). Therefore, Bayesian estimation is not beneficial to researchers who are committed to testing hypotheses using frequentist tools.

## Are violations of assumptions common in practice?

This lackluster image of statistical practice could be justifiable if violations of assumptions were not something occurring in practice, or if the OLS estimation method remained robust despite violated assumptions. Unfortunately, the evidence suggests to the contrary on both fronts. Initial studies of distributional characteristics focused on a set of educational achievement scores, demonstrating that high skewness and kurtosis are a common feature of these variables (*Lord, 1955*; *Cook, 1959*; *Burt, 1963*). In a landmark early meta-research, *Micceri (1989)* followed-up this work by studying the distributions of 440 educational and psychometric measures. *Micceri (1989)* found that almost all measures were significantly non-normal due to heavy tails, exponential levels of skewness, or multi-modality.

In the decades that followed, non-normal distributions have been recorded across different areas of psychological research. Skewness is now a well-documented characteristic in various time-dependent outcomes like reaction times (*Kranzler, 1992*; *Mewhort, Braun & Heathcote, 1992*; *Juhel, 1993*; *Reber, Alvarez & Squire, 1997*; *Leclaire, Osmon & Driscoll, 2020*), visual search and recognition tasks (*Hockley, 1984*; *Miyaoka, Iwamori & Miyaoka, 2018*), or eye-tracking responses (*Unsworth et al., 2011*), with exacerbated distribution tail in clinical populations (*Leth-Steensen, Elbaz & Douglas, 2000*; *Waschbusch, Sparkes & Northern Partners in Action for Child and Youth Services, 2003*; *Reckess et al., 2014*). A mixture of exponential and normal distribution is often reported as the most fitting distributional family for these measures (*Hockley, 1984*; *Juhel, 1993*; *Reber, Alvarez & Squire, 1997*; *Leclaire, Osmon & Driscoll, 2020*). Additionally, *Haslbeck, Ryan & Dablander (2023)* found that multi-modality and high levels of skewness are common in emotion time-series, while *Ho & Yu (2015)* confirmed the presence of non-normality in educational scale scores in a conceptual replication of Micceri's work. Focusing on broader psychological measures, *Blanca et al. (2013)* found that skewness and kurtosis are common, although they are not as extreme as originally reported by *Micceri (1989)*. *Cain, Zhang & Yuan (2017)* extends this work to multivariate designs. These studies have a key limitation, in that they focus on the distribution of raw variables. As noted above, the assumption of normality refers to the distribution of the model errors, not the variables included in the model. However, when studying the residuals extracted from OLS models, *Sladekova & Field (2024c)* found similar levels of skewness and kurtosis as those reported by *Blanca et al. (2013)*. In scenarios where distributions are non-normal, the distributional families researchers should expect to find most often are gamma, negative binomial, multinomial, binomial, log-normal, exponential, and Poisson, respectively (*Bono et al., 2017*). The only case where the normal distribution is a reasonable expectation seem to be the within-person T-scores of cognitive measures in non-clinical populations, which tend to be symmetrical with only some measures showing high kurtosis (*Buchholz et al., 2024*). Outside of this context, assuming normal distribution is, at best, wishful thinking. It could be argued that the non-normal distributions are not really a cause for concern—as we outlined above, we can often assume normal sampling distributions due to the central limit theorem, as long

as the sample size is large enough. While some psychological fields report average sample sizes in studies to be over 100 (*Fraley & Vazire, 2014*; *Reardon et al., 2019*; *Sassenberg & Ditrich, 2019*; *Fraley et al., 2022*), others lag behind (*Holmes, 1983*; *Marszalek et al., 2011*; *Hussey, 2023*), with samples sizes in some areas being as low as 10 (*Schrimp et al., 2022*). We elaborate this discussion below with reference to statistical power.

Likewise, non-constant variance is common in published research. *Grissom (2000)* reviews a range of clinical outcomes and illustrates how heteroscedastic variance can be a direct by-product of study design in psychological research. In a review of published reports, *Wilcox (1987)* identified group variance ratios as large as 16—that is, the largest variance among the groups being compared was 16 times that of the smallest variance. In a more recent research investigating heterogeneity of variance in a range of factorial designs, *Ruscio & Roche (2012)* found that variance ratio at the 50th percentile of the samples was 2.74, going up to 5.10 in the 90th percentile for all samples, and up to 9.43 for designs with at least four groups. Reviewing both published and unpublished work, *Sladekova & Field (2024c)* reports similar findings for residual variances while also noting heteroscedasticity in designs with continuous predictors.

## Are OLS models robust to violated assumptions?

Researchers therefore routinely neglect assumption checks even though they should realistically expect their violations in practice. This discrepancy is not entirely surprising—if our goal is to answer the question "Are OLS-based models robust to violations of assumptions?" with a simple yes or no, the messaging scattered in the past five decades of the robustness literature can appear mixed. If, however, we consider the effects violations of different assumptions on specific metrics of robustness, the general message is more consistent.

Perhaps the most influential work on the effects of violated assumptions on the ANOVA *F*-test is a review by *Glass, Peckham & Sanders (1972)*. In the rare instances where researchers report violated assumptions, Glass et al.'s work is often cited in support of researchers' decision not to take any remedial action. One of the conclusions presented in the review was the following: the rate of false positives of the *F*-test is unaffected by skewness. This conclusion has been consistently supported in further reviews and simulation studies—as long as skewness is the only concern, false positives will remain close to nominal 5% error rate (*Harwell et al., 1992*; *Schminder et al., 2010*; *Liu, 2015*; *Blanca et al., 2017*; *Yang, Tu & Chen, 2019*; *Delacre et al., 2019*). False positives will, however, be inflated in the presence of heteroscedasticity (*Glass, Peckham & Sanders, 1972*; *Rogan & Keselman, 1977*; *Harwell et al., 1992*; *Hsiung & Olejnik, 1996*; *Moder, 2010*; *Blanca et al., 2018*; *Nguyen et al., 2019*; *Yang, Tu & Chen, 2019*) and the effects on the analysis can compound when heteroscedasticity is combined with non-normal errors (*Delacre et al., 2019*; *Sladekova & Field, 2024a*). In factorial designs, the detrimental effects on false positives are further exacerbated when the group sample sizes are unequal (*Lantz, 2013*; *Delacre et al., 2019*)—a situation that often occurs in practice (*Sladekova & Field, 2024c*).

In a frequentist framework, an estimator's ability to control the rate of false positives is instrumental and should be the starting point in evaluating robustness if the goal is to

test the null hypothesis. Other metrics of robustness which have come to the forefront of considerations for statistical analyses only in the past couple of decades, include statistical power, estimation accuracy, and the coverage of confidence intervals (*Cumming, 2014*). Power is the probability of correctly rejecting a false null hypothesis. Informally, it is the probability of detecting an effect of a specified magnitude as statistically significant at a given alpha level (*Cohen, 2013*), and it is a crucial aspect of the robustness of OLS models. Both skewness and high kurtosis can reduce statistical power of OLS models below the recommended threshold of 80% (*Glass, Peckham & Sanders, 1972*; *Delacre et al., 2019*; *Nwobi & Akanno, 2021*; *Kim & Li, 2023*; *Sladekova & Field, 2024a*)—this fact is also highlighted in *Glass, Peckham & Sanders (1972)* but is often overlooked by researchers. Kurtosis produces distributions with heavier than normal tails—sampling from these distributions also reduces the coverage of confidence intervals (*Kim & Li, 2023*), and increases the probability of encountering outliers and influential cases, which can bias the parameter estimates (*Kim & Li, 2023*; *Sladekova & Field, 2024a*). Power plays an important role in the recent efforts to improve the credibility of psychology as a science (*Open Science Collaboration, 2015*; *Chambers, 2017*; *Vazire, 2018*), because inadequate power combined with poor control over false positives can alter the landscape of the available research findings. Studies with low power are less likely to reach statistical significance and are therefore less likely to be published (*Rosenthal, 1979*; *Sterling, Rosenbaum & Weinkam, 1995*; *Fanelli, 2010a*). Conversely, underpowered research that does reach statistical significance is more likely to reflect a 'lucky sample' and is less likely to replicate, but will stand a greater chance of becoming part of the published record (*Sterne, Gavaghan & Egger, 2000*). Attempts to synthesise or reproduce the effects found in published literature are seriously undermined by this kind of publication bias.

Thus far, discussions around power analyses have focused on the process of specifying the right sample size for a given effect size in order to reach sufficient statistical power for a hypothesis test (*Cohen, 2013*). Historically, sample sizes collected in psychological research have been low. In a series of investigations, *Holmes (1979)*, *Holmes, Holmes & Fanning (1981)* and *Holmes (1983)* reported no changes in sample sizes between 1950 and 1970, with median samples of 55 across four areas of psychology (Abnormal Psychology, Applied Psychology, Developmental Psychology, and Experimental Psychology), while *Marszalek et al. (2011)* found the median sample size across these areas to be even lower ($n = 40$) when replicating *Holmes's (1983)* study three decades later. Since then, sample sizes have improved in some areas of psychology. *Sassenberg & Ditrich (2019)* reports doubling of sample sizes between 2009 and 2018, with the lowest increase being from 122 to 185. Clinical research has seen the largest increase, with average sample size around 180 and statistical power to detect a moderate effect just below 90% in 2019 (*Reardon et al., 2019*). Doubling of sample sizes has also been observed in social and personality psychology between 2011 and 2019 (*Fraley et al., 2022*), with average sample size of 104 reported in 2014 (*Fraley & Vazire, 2014*). Other areas are still lagging on improvements—in applied and experimental behavioral research, most studies report samples up to 10 (*Schrimp et al., 2022*), while *Hussey (2023)* reports an average sample of $n = 64$ in Implicit Relational Assessment Procedure research, translating into statistical power of about 34%.

Sample sizes have therefore been at the forefront of focus in response to wide-spread replicability failures (*Open Science Collaboration, 2015*). Issues that can further affect statistical power, like violated assumptions, have not been given much spotlight because when assumptions are considered, it is usually as an afterthought rather than something that is accounted for in advance.

## Violated assumptions as a source of analytic flexibility

The possibility of violated assumptions or the presence of outliers and influential cases should be accounted for during the preparation of an analytic plan. This can prove to be challenging in practice. Methods that rely on pre-defined cut-off points for decision making (like significance tests) are easy to plan for in advance, but, as highlighted above, they are often not an appropriate tool for the job. Other methods—like for example diagnostic plots of residuals and fitted values—might provide a more nuanced picture of the problems, but they also rely on the subjective judgement of the researcher and often highlight only the most blatant violations (*Hayes & Cai, 2007a*; *Darlington & Hayes, 2017*). Decisions based on these methods can only be made after the data have been observed, and are inevitably subject to bias (*Fanelli, 2009*; *Steegen et al., 2016*).

While data-driven decision making is not problematic in its own right and is often utilised, for example, as part of Bayesian estimation and hypothesis testing (*Kruschke & Liddell, 2018*; *McElreath, 2020*), it is at odds with null hypothesis significance testing and the use of *p*-values as long run probabilities. Any analytic decisions made after the data have been observed can invalidate the *p*-value associated with test statistic for the hypothesis. The *p*-value is conditional on the decisions made about the study design and the analytic strategy. If these decisions are made before the data are observed, the *p*-value remains valid as a long run probability and the finding is potentially replicable, as long as the steps of the original study are being reproduced. If the analytic decisions are made *after* observing the data, the *p*-value becomes conditional on those decisions. In such situations, the meaning of the *p*-value changes to the probability of detecting the effect of the observed or greater magnitude if the null hypothesis is true *given* a specific set of decisions following a specific line of reasoning based on specific characteristics of the sample (*Greenland et al., 2016*). Replicating a finding based on a *p*-value conditional on one researcher's reasoning might be an impossible undertaking. On the one hand, there is great amount of flexibility when it comes to performing statistical analyses even in the simplest scenarios. Studies of the researchers' analytic degrees of freedom (*Steegen et al., 2016*) show that multiple researchers will come to divergent conclusions when faced with the same data and the same hypothesis (*Silberzahn et al., 2018*; *Botvinik-Nezer, 2020*; *Bastiaansen et al., 2020*; *Breznau et al., 2022*). The present paper alone has so far outlined only a small number of methods that could be combined and applied in a variety of ways, leading to divergent analytic paths. On the other hand, there is no guarantee that the same type of assumption violation or the same degree of violation will be detected in the sample of the replication study, in which case a new set of sample-based decisions needs to be made.

Flexible analytic decision making based on sample characteristics is therefore problematic in its own right, and it has recently received increased attention as a factor potentially

contributing to the publication of spurious unreplicable effects (*Simmons, Nelson & Simonsohn, 2011*; *Wagenmakers et al., 2011*). The existence of publication bias favouring statistically significant results (*Ferguson & Heene, 2012*) coupled with the incentive structures in academia and the 'publish or perish' culture (*Fanelli, 2010b*) puts researchers under a great amount of pressure to produce 'publishable' results. As such their decision making during the analytic process after observing the data may become biased towards obtaining a statistically significant result, even in the absence of deliberate attempts to manipulate the data. The efforts to improve the credibility of psychology have resulted in a number of proposed methodological reforms to address this. One such reform was the introduction of preregistration (*Nosek et al., 2018*), where researchers submit their analytic plans into a time-stamped online repository prior to collecting or accessing the data, and follow through with this plan when performing the analysis. Registered reports as publication format take this idea further—researchers submit a detailed study protocol, including the analysis plan, for peer-review and acceptance in principle is granted before the results are known, as long as the protocol is followed (*Chambers et al., 2015*).

There's growing evidence that registered reports are successful at combating publication bias (*Scheel, Schijen & Lakens, 2021*), however, preregistering a realistic plan can prove challenging when we consider the complications arising with violations of OLS conditions. As discussed, current methods for detecting violations require data inspection and post hoc decision making, while the most commonly applied 'remedies' (rank-based methods and data transformations) transform the data into a form that no longer maps onto the original hypotheses, introducing unnecessary degrees of freedom into the analytic process.

An ideal solution would be to preregister an alternative estimator with consistent performance across a variety of error distributions which also retains interpretation of parameter estimates comparable to OLS estimates. While no single method that meets these conditions outperforms all possible alternatives in all situations, a class of methods known as *robust statistical methods* offers a range of estimators that outperform OLS in a vast majority of scenarios in terms of statistical power and accuracy of estimates, and can therefore complement the efforts to improve the credibility of findings in psychology.

## ROBUST STATISTICAL METHODS

### What are robust methods?

A statistical method can be considered robust if (1) it has adequate control over the rate of false positive findings, (2) it retains adequate power to detect a true effect under a variety of scenarios, (3) the point estimates produced by the method are not overly sensitive to outliers—*i.e.,* they remain unbiased (4) the estimates accurately describe the typical individual in the sample, and (5) the previous points hold true regardless of whether or not the assumptions of OLS are violated.

"Robust methods" is an umbrella term encompassing a wide variety of methods that meet these conditions. This section will focus on some examples of robust methods that can be split into two general categories—methods improving the estimation of standard errors and confidence intervals constructed around the original OLS point estimates, and methods

improving the estimation of point estimates as well as the estimates of standard errors and confidence intervals. The first category includes bootstrapping (*Efron & Tibshirani, 1993*) and heteroscedasticity-consistent standard errors (*Hayes & Cai, 2007a*), while the second category includes $M$-estimators (*Huber, 1964*), and robust trimming (*Yuen, 1974*; *Wilcox, 1998b*). These four methods represent only small selection of all available robust methods. In an ideal world, the researcher would select a method that is known for optimal performance given the problem at hand. In the real world, it is unreasonable to expect the researchers to have an in depth knowledge about the performance and application of every single method there is, and statisticians or methodologists who would be able to advise on the issue while also understanding the research context may not be readily available in most psychology departments (*Golinski & Cribbie, 2009*). We focus on the methods outlined above because they are fairly intuitive extensions of the standard methods, easily applied to common research designs in psychology, and guidance already exists for their application (*Hayes & Cai, 2007a*; *Wilcox, 2017*; *Field & Wilcox, 2017*). The following section provides an introduction to these methods, followed by a discussion of the benefits of their application.

## Robust standard errors and confidence intervals
### Bootstrapping

Bootstrapping is a re-sampling procedure that allows for the empirical estimation of standard errors (*Efron & Tibshirani, 1986*; *Efron & Tibshirani, 1993*). A bootstrap sample is collected by sampling with replacement from the current sample. For example, if we have a sample containing the values 1, 4, 3, 1, 8, we could, by a random selection, create a bootstrap sample of 1, 8, 1, 1, 4 or 3, 4, 3, 4, 1. Each bootstrap sample is the size of the original sample. Once we have a bootstrap sample, we can compute the the parameter estimate for this sample of this sample. This process is typically repeated about 1,000 times, resulting in an empirical distribution of sample estimates. We can then use this empirical sampling distribution to estimate the standard error and to construct a bootstrapped confidence interval by looking at the lower and upper limits of the middle percentage (typically 95%) of the sample means. This is called a percentile bootstrap.

Bootstrapping can be helpful for obtaining more accurate confidence intervals in small samples, or samples with heavy tails. In extremely small samples ($n < 20$) the performance of the bootstrap can deteriorate (*Canty, Davison & Hinkley, 1996*), however the confidence intervals will still be more accurate than intervals based on OLS standard errors (*Wilcox, 2010*; *Wilcox, 2017*). The percentile bootstrap described above is only one of many bootstrapping methods. There are other types with more optimal performance that extends to a wider variety of situations. The bias corrected and accelerated ($BC_a$) bootstrap outperforms other bootstrap methods in a number of respects. Overall, $BC_a$ tends to reach more accurate confidence intervals coverage probability (*i.e.,* the proportion of the simulated intervals that contain the true population value is close to the intended 95%) than the percentile bootstrap and the bias correction can account for skewness as well as heavy-tails (*Hall, 1988*; *Efron & Tibshirani, 1993*; *DiCiccio & Efron, 1996*). The main critique of the $BC_a$ bootstrap used to be that it is computationally intensive and does not offer advantage to simpler bootstrap methods when the variables are transformed for

normality prior to applying the bootstrap (*Canty, Davison & Hinkley, 1996*). However, modern computing power means that the $BC_a$ can be applied with relative ease to most situations and, as discussed, transforming the data can often do more harm than good (*Osborne, 2002*; *Grayson, 2004*; *Schmidt & Finan, 2018*).

### Heteroscedasticity-consistent standard errors

There are also robust estimators of standard errors known as heteroscedasticity-consistent standard errors (HCSE) estimators. This class of estimators work on the principle of estimating the covariance matrix based on the sample residuals in a way that does not assume homoscedasticity, allowing for valid inferences when the assumption is violated. *Hayes & Cai (2007a)* provide an introduction to the theory and application of HCSE estimators. As with bootstrap, several HCSE estimators are available, namely HC0 (*Eicker, 1967*; *Huber, 1967*; *White, 1980*), HC1 (*Hinkley, 1977*), HC2 (*MacKinnon & White, 1985*), HC3 (*Davidson & MacKinnon, 1993*), and HC4 (*Cribari-Neto & Lima, 2009*). The HC1–HC4 estimators represent the developments in the computational strategies of the original HC0 estimator to improve performance under a wider variety of scenarios. Overall, HC3 outperforms its predecessors when it comes to the accuracy of the confidence interval coverage probability (*Long & Ervin, 2000*; *Cribari-Neto & Zarkos, 2001*; *Cai & Hayes, 2008*), where the coverage of HC0-HC2 tends to be too small. HC4 is an extension of HC3 that also accounts for the presence of influential cases in the sample, and retains good coverage even when heavy-tailedness is combined with heteroscedasticity (*Cribari-Neto & Lima, 2009*). Conversely, none of these estimators have good coverage when heteroscedasticity is coupled with exponential levels of skewness (*Cribari-Neto & Lima, 2009*). Another drawback of these estimators is that their performance can be suboptimal in small sample sizes (∼25), especially in situations where homoscedasticity is met, resulting in a loss of power to detect a real effect (*Godfrey, 2006*; *Cribari-Neto & Lima, 2009*). With larger samples however ($n > 60$), there is no evidence that HC4 is negatively affected in the presence of homoscedasticity, unlike the previous versions of the HCSE estimators (*Godfrey, 2006*).

To sum up, the bootstrap is most beneficial in small sample sizes, particularly when the main concerns are skewness or heavy tails, whereas the HC4 estimator is useful in moderate to large sample sizes, and is able to account for heteroscedasticity, influential cases, as well as heavy-tails. A special situation worth noting is one where the form of heteroscedasticity is unknown—that is, the error distribution is not a function of the fitted values, but it is still not constant, and therefore not homoscedastic. This can happen when the model is misspecified and there are predictors missing from the model that are generating the heteroscedasticity (*Hayes & Cai, 2007a*). In such situations, bootstrapping as described above is not appropriate as the unknown pattern, by its definition, cannot be replicated by the bootstrapping re-sampling process (*Wu, 1986*). In such cases, the HC4 estimator coupled with so called wild bootstrap (*Liu, 1988*) shows the best performance compared to the its HC counterparts, alternative bootstraps, and the OLS error estimator (*Godfrey, 2006*; *Davidson & Flachaire, 2008*).

## Robust point estimates

A point estimate for the relationship between variables that we are studying should represent a typical individual in a given population as accurately as possible and produce standard error of size that is not detrimental to statistical power. No estimator beats OLS when sampling from a normal distribution. However as we have seen, such distributions are an exception rather than the rule in applied settings. OLS can be especially volatile in heavy-tailed distributions that are likely to contain outliers. The variance computed using OLS will be too large and therefore the power to detect a real statistically significant effect will be low. Additionally, OLS becomes biased when outliers are present. This is related to its low finite sample breakdown point, which is the proportion of extreme values in the sample that can cause the estimate to be biased in the direction of these values. Each estimator has its own breakdown point, and for OLS this value is $1/n$, $n$ being the sample size. In proportional terms, the value gets smaller as the sample size increases, therefore large sample sizes are not protected from the effects of outliers.

### M-estimators

Among the robust methods that show substantially better performance than the mean (which is an OLS estimator) are the trimmed mean and $M$-estimators. $M$-estimators are among the robust methods that show substantially better performance in situations where OLS becomes biased or produces non-optimal estimates. During $M$-estimation, extreme observations are weighted down to lessen their impact on the point estimate. Which observations are weighted down and to what extent is determined algorithmically by the computer based on the properties of the model in question. The researcher can however select a function according to which the down-weighting should be applied.

One example of a weight function is Huber's $\psi$ (*Huber, 1964*), which applies smaller weights to observations with large residuals, where the residuals that are beyond a specific point are given a weight of zero—this the equivalent of the scores being trimmed off and not weighing in on the computation of the point estimate. Figure 3A illustrates one form Huber's $\psi$ can take. Observations with residuals between −1.2 and +1.2 are assigned the same weights as they would have in and OLS estimation, whereas the observations with the residuals outside of this boundary are given a weight of zero—this is represented by the horizontal lines at the tails of the weight function. As such, $M$-estimation is always completed in two or more steps. First, an OLS model is fitted and the weights are determined based on the residuals in this model, then an $M$-estimate is computed. The process is iteratively repeated until the model converges (*Susanti et al., 2014*). The breakdown point of $M$-estimators is 0.5, meaning that 50% of scores can be at the extreme ends of the curve and the point estimate will remain at the same location. This is a maximum possible breakdown point an estimator can have. As a result of the weighting procedure, the standard errors in the model that uses $M$-estimation are smaller than they would be in an OLS model when sampling from heavy tailed distribution, thus increasing the power of the associated statistical significance tests (*Wilcox, 2010*).

$M$-estimators using Huber's $\psi$ for weighting can however be inefficient under heteroscedasticity and when there are influential cases in the sample (*Croux, Dhaene &*

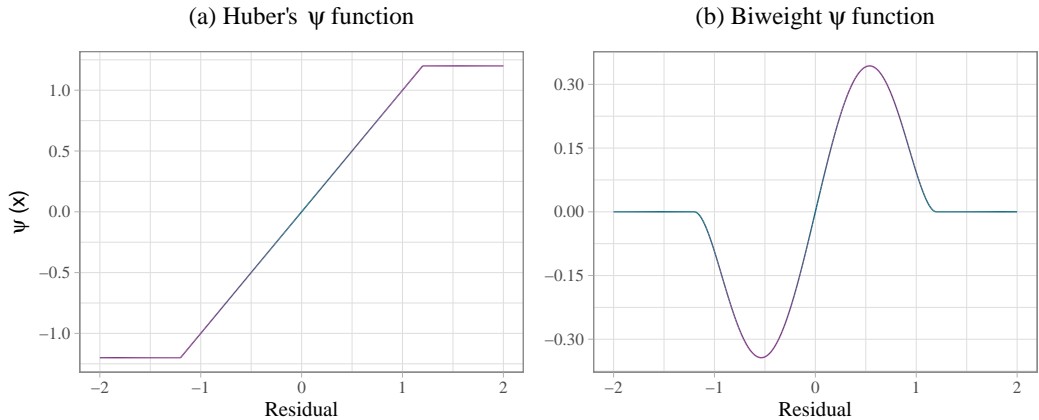

**Figure 3** (A) Huber's $\psi$ weight function, defined as $\phi(x_i) = k \times sign(x_i)$ if $|x_i| \geq k$ and as $\phi(x_i) = x_i$ if $|x_i| < k$, for $k = 1.2$; (B) Tukey's redescending biweight $\psi$ function , defined as $\phi(x_i) = x_i \times (k^2 - x_i^2)^2$ if $|x_i| < k$ and as $\phi(x_i) = 0$ if $|x_i| \geq k$, for $k = 1.2$, $x$ being the value of the residual. These functions are used to determine the weights in $M$-estimation, while the value of $k$ is selected using an iterative algorithm.

*Hoorelbeke, 2003*; *Toka & Cetın, 2011*). Extensions of this estimator have been derived over the years to deal with these issues. We focus on the $S$-estimator, $MM$-estimator, and $DAS$-estimator, although other versions of the $M$-estimator exist (*Wilcox, 2017*). Each of these estimators builds on the previous version by iterating through the original steps and then applying a unique adjustment. The $S$-estimator affects the variance and aims to minimise the dispersion of the residuals around the $M$-estimate (*Rousseuw & Yohai, 1984*; *Toka & Cetın, 2011*). The $MM$-estimator (*Yohai, 1987*) applies an alternative redescending function (often Tukey's biweight function, Fig. 3B) to weigh the residuals and re-estimate the point estimate originally produced by $M$-estimation and the variance estimate produced by $S$-estimation in the previous steps. The weakness of the $MM$-estimator is that it becomes inefficient in designs with small samples (below 20), but also in larger samples if the ratio of the number of predictors in the model to the number of observations ($p/n$ ratio) is greater than 1/10. In other words, if there are less than 10 observations for each predictor in the model, the estimate suffers. High $p/n$ ratio also causes bias in the estimation of variance (*Croux, Dhaene & Hoorelbeke, 2003*). The Design Adaptive Scale estimator (DAS; *Koller & Stahel, 2011*) was developed to correct this bias and lack of efficiency, and can do so successfully for ratios of up to $p/n = 1/3$. The function applied by this estimator is also less steep than the typical the biweight function typically applied in $MM$-estimation, which accounts for overly aggressive down-weighting of observations as outliers in heteroscedastic models.

### Trimming

Trimming can be thought of as a specific case of $M$-estimation developed for GLM designs that compare group means. Given that the(arithmetic) mean is itself a least square estimator, same limitations as outlined above for more general forms of OLS apply. Most researchers are likely familiar with the concept of trimming and the trimmed mean, but not

necessarily with the benefits to estimation this procedure provides. In essence, trimming is cutting away a proportion of data points at the tails of the sample distribution. A 5% trimmed mean is a mean computed for a sample where lower and upper 5% of the scores are not included in the computation. A trim of about 50% results in a median. While the median is resistant to the presence of outliers, it can be very inefficient when it comes to estimates of variance, and comparisons based on this measure can have low power (*Wilcox, 1998a*; *Wilcox, 1998b*). Researchers might be uncomfortable with the idea of trimming away fairly large proportions of data, and this intuition is not entirely misguided. The crucial point here is that the process of obtaining a trimmed mean and conducting statistical comparisons based on the trimmed mean is not simply discarding 20% of the data on each end of the distribution and then applying OLS estimation to obtain the standard errors and conduct statistical significance tests. Computing a trimmed mean involves ordering the observations from largest to smallest. As such, the observations are no longer independent, and the estimates of variance become inaccurate. *Wilcox (2010)* and *Wilcox (1998b)* discuss this issue in more detail. Note that manual al of outliers faces the same problem, and an OLS model fitted to a sample after the outlier removal will also produce inaccurate standard errors (*Field & Wilcox, 2017*).

Statistical procedures for computing accurate variance and comparing the point estimates have been derived (*Yuen, 1974*; *Wilcox, 1998b*; *Wilcox, 2010*; *Wilcox, 2017*), and these methods are known to perform well under non-normality and heteroscedasticity. Determining the right amount of trimming can vary depending on the situation, however 20% trimmed mean outperforms least-squares mean in terms of power and accuracy in a wide variety of situations (*Rosenberger & Gasko, 1983*; *Lix & Keselman, 1998*; *Wilcox, 2017*). *Rosenberger & Gasko (1983)* note that for small sample below 20 with heavy tails, 25% trim is recommended, whereas a 20% trim will suffice in samples larger than that. Under exact normality, the trimmed mean can suffer a small loss of power, however show great advantage when normality cannot be assumed (*Yuen, 1974*; *Wilcox, 1998b*; *Lix & Keselman, 1998*; *Keselman et al., 2002*). *Wilcox (2010)* notes that trimmed means might be more suitable in small samples when comparing differences between groups, whereas $M$-estimators have more utility in regression designs with larger samples (however note that coefficients in dummy-coded regression designs can also represent the differences between groups, as both designs are part of the GLM framework (*Cohen, 1968*; *Field, 2024*).

## IMPROVING TRANSPARENCY AND CREDIBILITY OF FINDINGS WITH ROBUST METHODS AND SENSITIVITY ANALYSIS

As highlighted above, violations of OLS conditions are common in applied settings. In such scenarios, the parameter estimates or the hypothesis tests produced by OLS may encourage misleading conclusions. Despite this, OLS is often applied uncritically as the default option. This may be due to researchers' unfamiliarity with robust methods (*Sladekova & Field, 2024b*) or simply the result of a contagion effect—researchers fall back to the familiar OSF because (a) it's what they've always done (b) it's what their peers, mentors or
supervisors have always done and thus recommend doing again or (c) it's what they keep finding in journals in which they wish to publish their work.

It is therefore somewhat understandable that OLS is often perceived as the "safe" option that may lessen potential tensions between the authors and their collaborators, supervisors, or peer-reviewers. However, if the purpose of running a statistical analysis is to further the discipline's knowledge with scientific discovery, planning and preregistering analyses that can deal with skewed and heteroscedastic error distributions with outliers should be a priority rather than an afterthought. The robust methods outlined above have been evaluated under a wide variety of conditions. One conclusion that remains evident across studies is that in most situations researchers are likely to encounter in practice, robust methods are the superior choice to OLS, providing more accurate parameter estimates, unbiased significance tests and better statistical power (*Yuen, 1974*; *Rosenberger & Gasko, 1983*; *Wu, 1986*; *Liu, 1988*; *Wilcox, 1998a*; *Croux, Dhaene & Hoorelbeke, 2003*; *e.g.*, *Toka & Cetın, 2011*; *Koller & Stahel, 2011*; *Sladekova & Field, 2024a*). Unlike classic non-parametric tests, robust methods are adaptable to a range of designs commonly used in psychological research. This means that they can adequately supplement or indeed replace any model that was originally designed for OLS-based hypothesis testing. In a recent review, robust methods were highlighted as a tool that can aid replication due to their consistent performance in models with heavy-tailed error distributions (*Yuan & Gome, 2021*). Nevertheless, robust methods remain under-utilised (*Sladekova & Field, 2024b*; *Sladekova, Poupa & Field, 2024*).

Although robust methods consistently outperform OLS, they differ in performance when compared to each other. Therefore, the choice of best-suited robust method is important to maximise the benefits of their application. Some circumstances that affect performance are under the researcher's control and can be planned for in advance. These include the sample size, the ratio of sample sizes between groups, and aspects of model design such as type and number of predictors.

Some methods will, however, produce parameter estimates that describe the typical individual from the population with more accuracy compared to others if the error distributions in the population model are asymmetrical due to skewness and heteroscedasticity. Consider the case of a population mean calculated for a skewed distribution as shown on Fig. 4. When sampling randomly from this population, the means of individual samples would eventually converge on the population mean in a sampling distribution. The mean would therefore not be biased—there's no systematic over- or under-estimation of the population mean—but it also wouldn't be the most accurate representation of a typical individual from that population. If finding a value for such an individual is our goal[2] then a robust estimator less affected by asymmetry (like the trimmed mean or the *MM*-estimator) is a better choice. We summarised the performance of the methods introduced above in detail in a related simulation study (*Sladekova & Field, 2024a*). Here we provide some practical guidance on how researchers can use robust

[2]In most conventional scenarios, this will be the case, however see *Ng & Cribbie (2017)* who makes the case for characterising the whole distribution using generalised linear models instead of minimizing the effects of tails using robust models.

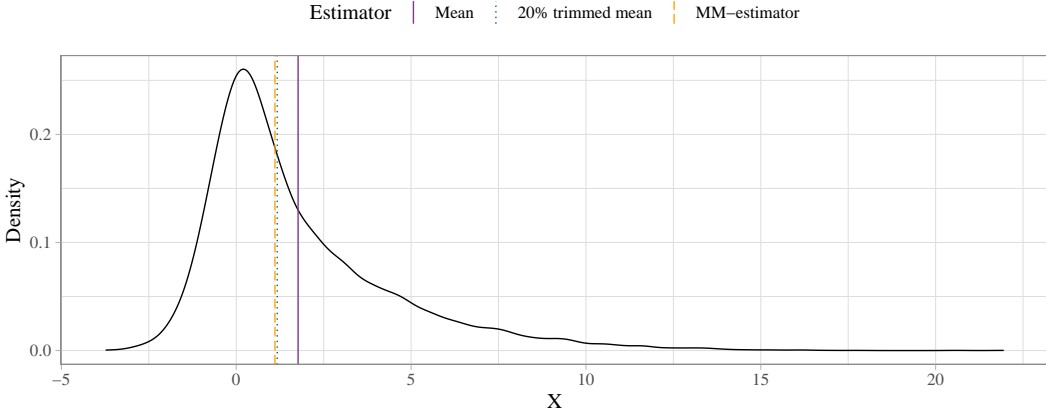

**Figure 4** Estimation accuracy of robust estimators compared to the mean in a skewed population distribution (illustrative example).

methods in a sensitivity analysis while also preregistering their plan ahead of the analysis and reducing problematic degrees of freedom.

Sensitivity analysis is the process of determining how sensitive our results are to the choice of our model and the assumptions we are making about the population or the data-generating process. At the extreme end, it can take the form of a multiverse analysis, where a multitude of analytic paths are explored and variability or stability of estimates from the different paths is considered when the results are being interpreted (*Steegen et al., 2016*; *Girardi et al., 2024*). In Bayesian estimation, parameter estimates and Bayes factors are often examined under different prior distributions to examine the effects that priors have on the conclusions drawn from the data (*McElreath, 2020*). An important feature of a sensitivity analysis is transparency—that is, the results of all different analyses are made available for inspection and the authors discuss the implications of any diverging results.

If researchers wish to avoid or minimise data-driven decisions in a frequentist analysis, a sensitivity analysis with an appropriate robust method or methods can be preregistered. Researchers would need to identify a set of error distributions that can be plausibly expected for the effect they are studying. As discussed, distributions associated with some variables are well documented. For example, reaction times often generate right skewed error distributions (*Whelan, 2008*), while some clinical measures can create heteroscedasticity (*Grissom, 2000*). For researchers in fields with less thoroughly documented distributional characteristics, we provide an overview of most common values for skewness, kurtosis, heteroscedasticity, as well as other relevant metrics in a study examining over 500 models reported in both published and unpublished psychology papers (*Sladekova & Field, 2024c*). These values can be used as a proxy for preregistering a sensitivity analysis of estimators that can deal with the "typical", "worst" and "best" case scenarios. We evaluated the robust methods introduced here under these scenarios in a recent simulation study (*Sladekova & Field, 2024a*), based on which we can make the following recommendations:

- If the normal errors with constant variance can be assumed, or if light-tailed symmetrical and homoscedastic distributions free of outliers are expected, OLS remains the best choice.
- In scenarios where skewed and heteroscedastic error distributions or outliers are expected, OLS, bootstrapping, and HCSE should be avoided if the goal is to estimate the population value accurately. HCSEs do not affect the parameter estimates, while the repeated bootstrap sampling from a population with a skewed distribution will result in the empirical sampling distribution converging on an inaccurate value.
- For between-subjects or cross-sectional designs with skewed and heteroscedastic distributions researchers can preregister either the *MM*-estimator or the *DAS*-estimator. Both estimators perform well in designs with categorical or continuous predictors. The *MM*-estimator may fail to converge in up to 6% of cases if the sample size is insufficient relative to the number of predictors in the model—the *DAS*- estimator can be preregistered as a contingency plan for this possibility. The *DAS*-estimator, on the other hand, should be avoided if the group sample sizes are too imbalanced (ratio of the largest group to the smallest is more than 2:1)
- For repeated-measures designs, 20% robust trimming can be beneficial for skewed and heteroscedastic error distributions, while 20% trimming combined with the *t*-bootstrap for the estimation of confidence intervals is more useful with mixed designs if the group sample sizes are unbalanced.
- HCSE and either bootstrapping method remain a valid option if the error distribution is symmetrical. They are especially advantageous in designs with heteroscedastic categorical predictors with heteroscedasticity. Bootstrapping should be prioritised in small samples with balanced sample sizes across groups, while HCSE are better at handling unbalanced designs, as well as extreme levels of heteroscedasticity (variance ratios above 4).

Once the estimates from the different models are obtained researchers can interrogate them in the context of the assumptions that can be made about the error distribution. For example, if we pre-registered an OLS model, a BCa bootstrap, and an *MM*-estimator, the OLS estimates will be most accurate if we assume normal and homoscedastic errors with no outliers. If we only managed to collected a relatively small sample ($\sim$70) and we assume symmetric errors with heteroscedasticity, the OLS estimate and the bootstrapped estimate will likely converge, however the bootstrapped statistical tests will be better powered and we could therefore observe a discrepancy in significance tests. If we assume skewed heteroscedastic errors, the most accurate parameter estimates will be provided by the *MM*-estimator. Researchers should especially pay attention if the parameter estimates for the *MM*-estimator and the other two estimators diverge, as this could indicate that the model errors are asymmetrical or that there are outliers in the sample. *Field & Wilcox (2017)* suggests that in such scenarios, the *MM*-estimate should be interpreted instead.

For some researchers, this approach might seem (understandably) unsatisfactory— instead of a single result they would typically obtain, they are now left with three or more potentially contradicting estimates and significance tests. However, uncertainty is a key component of statistical research and this is especially true with frequentist

concepts like *p*-values and confidence intervals. It is important to express this uncertainty transparently, especially when it arises as a result of factors beyond the researcher's control, such characteristics of the population that is being studied. This further highlights the need for replication, open data sharing and cumulative approach to science, so that the variables psychology researchers work with and their impacts on error distributions are systematically documented in a way that allows researchers to make realistic analytic plans instead of assuming normal distributions and hoping for the best.

In this paper, we presented an approach that combines recently introduced open science tools like pre-registration with robust methods to allow researchers to estimate values that accurately describe the typical individual in the population. Notably, other approaches exist and researchers should be aware that robust estimators are not suitable if the researcher wishes to be able to make predictions about the individuals at the tails of the distribution. In such situations, generalised or generalizable linear models are a more appropriate tool. *Ng & Cribbie (2017)* provide an accessible introduction to this issue.

Finally, researchers should note that the application of robust methods is not a substitute for thorough data checks that could uncover issues going beyond the violations of OLS conditions. Among other things, this could include data entry and processing errors, careless responding, or extreme cases not well predicted by the model that warrant further investigation. Additionally, while robust methods can adequately deal with violated assumptions, they do not address estimation problems that could arise from other sources of bias, like missing data, model misspecification due to omitted variables, confounding, or sampling bias.

## LIMITATIONS

We aimed to provide a synthesised overview of the literature available across several research areas however this endeavour has had several challenges. Studies reviewing statistical practice often rely on the information available in published papers, however this remains, by and large, at the discretion of the researchers producing those papers. An absence of assumptions checks in a published report does not guarantee that an assumption check has not been carried out. Conversely, a report of "satisfied" statistical assumptions does not guarantee the assumption were realistically met in practice or checked with appropriate methods. Although many journals now put more emphasis on open science and adherence to standard reporting guidelines, this does not guarantee sufficient transparency (*Wicherts, Bakker & Molenaar, 2011*) necessary to fully understand statistical practice and its effects on the credibility of findings in the field. We supplemented this gap by including empirical papers examining researchers' practice in self-reports or experiments, however the literature of this kind remains sparse in the context of OLS assumptions.

In general, simulation studies suffer from a lack of methodological consistency. The conditions simulated for evaluating model performance—like levels of skewness, kurtosis, or heteroscedasticity—can range widely from one study to another, often without satisfactory justification for how and why specific distributions had been selected.

Simulations that draw directly on conditions found in real data are an exception rather than the rule, and even then methodologists are limited to the kind of data researchers are willing to share in the first place. *Luijken et al. (2024)* also notes that simulation studies frequently lack sufficient detail to enable reproducibility. This is means that each new simulation study evaluating the performance of an estimator not only implements changes to simulated conditions suitable for answering the research questions, but might also need to re-invent the wheel and implement an entirely different algorithm from scratch for generating random samples. This creates a challenge for comparing the performance of various estimators unless they are evaluated in a single study. Incidentally, this is why the conclusion that OLS is an inferior choice compared to robust estimators when assumptions are violated is rarely up for a debate—a single study will typically pit one or more robust estimators against the OLS and can therefore reliably compare the estimators' performance in their simulated ecosystem of distributions. However, a study evaluating the performance of bootstrapping will not necessarily also evaluate the performance of HC4 or $M$-estimators, which makes the comparison of different robust methods to each other challenging and prevents us from definitively recommending one robust method over others. Any conclusions and guidelines presented in this paper were made with these limitations in mind.

## CONCLUSION

We introduced several robust methods and highlighted the issues that current practice associated with OLS estimation can have for broader replicability and credibility of psychological findings. Of all the methods discussed, OLS is the least equipped to deal with data typically found in psychology research, yet it remains the most frequently applied method. When the assumptions of OLS are violated, the estimator loses efficiency and statistical power, and can become biased with a single extreme observation present in the sample. Robust methods remain unbiased and accurate in the presence of outliers, and retain statistical power in common applied settings where the OLS models fail. Unlike tests on transformed variables or the classic non-parametric tests (Wilcoxon-Mann-Whitney or Kruskal-Wallis test), robust methods keep the model properties interpretable and can be flexibly applied to more complex designs.

We have argued that a pre-specified sensitivity analysis using robust methods is preferable to a *post-hoc* application of countermeasures after the violations of assumptions are detected. Not only can decision-making based on statistical tests of assumptions increase the rate of false positive findings, but it can also introduce bias into the analytic process and render the findings impossible to replicate. The use of robust methods reduces the need for *post-hoc* decision making, allows the researchers to preregister and carry out more realistic analysis plans, while remaining transparent about the impact that unknown error distributions can have on their conclusions. Robust methods are not a quick-fix solution to inadequate sampling or poor methodological choices, and no single method outperforms others in all possible situations. However in situations that the researchers are likely to encounter in applied settings, robust methods offer an advantage in the accuracy

of estimates and the power of statistical tests when compared to OLS estimation. Routine application of robust methods could contribute to improving the credibility of psychology as a science by increasing the robustness of statistical findings as well as enabling more transparent and reproducible practice.

## ACKNOWLEDGEMENTS

We would like to thank Dr. Alyssa Counsell and Dr. Dominique Makowski for their feedback on the early version of this manuscript.

### Funding

This article was written as part of doctoral research funded by the Economic and Social Research Council [ES/P00072X/1]. The funders had no role in study design, data collection and analysis, decision to publish, or preparation of the manuscript.

### Grant Disclosures

The following grant information was disclosed by the authors:
Economic and Social Research Council: ES/P00072X/1.

### Competing Interests

The authors declare there are no competing interests.

### Author Contributions

- Martina Sladekova conceived and designed the experiments, performed the experiments, analyzed the data, prepared figures and/or tables, authored or reviewed drafts of the article, and approved the final draft.
- Andy P. Field conceived and designed the experiments, authored or reviewed drafts of the article, and approved the final draft.

### Data Availability

This is a literature review.

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
