# Peer review of "Robust statistical methods and the credibility movement of psychological science"

_PeerJ, doi:10.7717/peerj.20043_

## Round 0.1 · original submission · Major Revisions

· Academic Editor

Major Revisions

Both Reviewers had concerns about the search string. Given that this is a literature review, this is a crucial aspect, and it cannot be overlooked. Please address the concerns of both reviewers, even by carrying out a new search.
Both Reviewers pointed to a general superficiality in the discussion the topics. Given the crucial (in my view) point of the manuscript, a more comprehensive, detailed, and formal discussion of the topics is needed. Moreover, as R2 did, I noticed some inconsistencies in the terminology that you have used throughout the manuscript. Please, you have to be consistent, otherwise, it is extremely difficult to understand the topic.

I truly believe that this manuscript is beneficial for the social sciences. However, the issues raised by the reviewers need to be carefully addressed.

·

Basic reporting

No comment, i.e. this article clearly meets the required basic reporting standards that are outlined.

Experimental design

While I believe this article meets the required study design standards that are outlined, I would make a brief comment on the search methodology.

The authors used major article databases (Scopus, Web of Science, etc) to search for articles in psychology using specific search terms to address three questions:
* Do researchers attend to assumptions?
* How common are violations?
* How do OLS models perform when assumptions are violated?

While all of that is highly appropriate, I did wonder about the search terms used. For example, for the first question, the string "assumption* AND (model* OR statistic* OR linear OR regress*) AND (check* OR test* OR assess*)" was used. Not that I see any particular problem with that or the other search terms, it is just not obvious to me why that particular search term and not others were used. Were others tried and found to be inadequate? Can we be confident that this is the best search term to find all the relevant articles? And how many irrelevant articles were found and how were these filtered out?

I would prefer, therefore, some elaboration about these search terms and their justification and their adequacy to find all only the relevant articles.

Validity of the findings

Again, while I believe the article meets the required standard for validity of findings, I do have some comments.

The article strongly recommends improved statistical practices to deal with the inadequacies of GLMs, specifically the widespread use of GLMs in contexts and problems where there are major violations of assumptions. However, I felt the coverage of the robust estimators was quite brief and lacked sufficient detail to be of practical value. For example, the descriptions in the subsection What Are Robust Methods? were all essentially just brief descriptions of quite complex topics. Of course, this article can not be expected to provide a comprehensive introduction, which would require a book length treatment, but given how it seems that one of the major points of this article is to advocate alternatives to standard default practices, I think more needs to be said than is currently.

There is a particular focus on M estimators over alternative robust methods so perhaps this one in particular should be elaborated. Perhaps some coverage explaining in more detail what M estimators are in general, paralleling the section that introduced GLMs and OLS estimators, and then perhaps some mention of how M estimators can be used in practice, e.g. in R.

As an additional comment, I wonder if something more should be said about Bayesian methods. To be clear, personally I think Bayesian methods can, and in fact are, used to deal with many of the major problems elaborated in this paper. On the other hand, this is not my paper and I do not want to do the annoying you-must-write-the-article-I-would-write thing that some reviewers do. I don't see anything wrong with this article sticking to the general frequentist framework and considering other methods, i.e the robust methods mentioned above, that are available though not commonly appreciated in this framework. I don't see any necessity to turn this paper into advocacy of Bayesian methods. On the other hand, Bayesian methods are becoming widely used and so others readers may wonder about whether or how Bayesian methods can be used to address the problems identified in this paper. As it stands, there is almost no mention of Bayesian methods. I think something should be said to explain that Bayesian methods are an alternative approach, even if that alternative approach is not the focus of this paper. I think otherwise there might be some "What about Bayes? How does that fit in here?" type questions on readers' minds.

Additional comments

I think this is a very valuable article. The authors are pointing out that the default methods in Psychology and related fields are based on assumptions that often not met in practice and when they are not met in practice, they can in fact lead to misleading results. I think this point needs to be repeatedly made because while lip service is paid to model assumptions and model mis-specification, in practice a default set of tools are used uncritically and almost unconsciously. These methods are used in practice because they are used in practice (e.g. "everyone else uses repeated measures ANOVA for reaction time analysis, so that's what we will use too; no questions asked") and are often uncritically accepted in peer review for the exact same reason. And the authors also show how assumptions are routinely not checked or reported, or are misunderstood, or remedied by methods that can make matters worse.

An additional comment, and this comment is drifting into the realm of pedantry, is about the distinction between the general linear model and the normal linear (aka linear-Gaussian) model, and also between ordinary least squares and maximum likelihood estimators. In the subsection beginning on line 110, what is described there, strictly (pedantically) speaking is a normal linear model, and because OLS estimation is maximum likelihood estimation, the subsection beginning on line 143 is about the OLS and maximum likelihood estimators (they are, in this context, the same thing) and their sampling distribution.

I don't see anything wrong with this approach. In practice, the general linear model is almost always a normal linear model. I just think that maybe something needs to be said to make that point more explicit. For example, the definition of general linear models could be made, and then it is stated that the epsilon_i are routinely assumed to be iid normal, and this normality assumption is necessarily required in all the widely used methods like ANOVA, multiple regression etc etc, and so it is that model per se that is the focus of this article rather than the technically, though not necessarily practically, more general topic of general linear models.

Reviewer 2 ·

Basic reporting

The manuscript is written in clear and professional English.

Although I am not a grammar expert, I noticed some ambiguous sentences and occasional misuse of statistical terms throughout the text. Some examples:

- Line 12: "which produces unbiased and optimal parameter estimates,"
- "This method produces unbiased and optimal point estimates of parameters"
- "tests associated with these estimates will be unbiased and optimal"
- "is not optimal is not necessarily biased, but has larger variance"
-: "produced by a model containing outliers will still be optimal"
- "in order to reach optimal statistical power for a hypothesis test"
- "the population with more accuracy compared to others if the error distributions"
- "if the goal is to estimate the population value accurately."
- "most important conditions for an unbiased model"
\end{enumerate}

The Introduction adequately introduces the subject and clearly articulates the motivation. However, further clarification is needed regarding certain additional assumptions, which are detailed in the attached pdf in the "Main comments" section.

The manuscript aligns with the scope of the journal by introducing a manual to deal with assumption violations and proposing the inclusion of model assumptions in the pre-registration step.
It would be useful to add a section that directs readers to additional resources, such as tutorials, courses, or books that delve deeper into robust statistical methods, it could make the manuscript a valuable resource for both novices and advanced researchers in the field.


Overall, the manuscript is well-written, and I believe it could make a significant contribution to PeerJ.

Experimental design

The manuscript effectively describes the criteria for selecting studies included in the review and the process of data extraction, which are essential for ensuring transparency and reproducibility in literature reviews.

However, there seems to be a discrepancy in how the manuscript handles the concepts of outliers and influential observations. While these topics are discussed in the context of their impact on statistical analyses, they are not reflected in the keywords used for querying the literature in question (3). Including terms such as "bias," "outliers," and "influential observations" as part of the search strategy would enhance the thoroughness of the literature review.

Validity of the findings

The discussion on the effectiveness of these methods under various assumption violations appears is based on a well-structured literature review. However, the authors could improve the discussion about assumption violations and robust methods to deal with these. For example, if we have no normality.. the estimator will be ... and the approach XX resolve this...

Moreover, the authors should ensure that the conclusions drawn are cautious and reflect the limitations of the studies reviewed. It would be beneficial for the manuscript to include a section on the potential biases in the literature and how they might affect the validity of the review’s conclusions. This approach will strengthen the credibility of the findings and provide readers with a clearer understanding of the contexts in which these robust methods are most effective.

Additional comments

Please refer to the attached pdf

Annotated reviews are not available for download in order to protect the identity of reviewers who chose to remain anonymous.

---

## Round 0.2 · Minor Revisions

· Academic Editor

Minor Revisions

Dear Authors,

I have now received the revisions of both Reviewers, who are quite satisfied with your work. However, there are still some minor concerns, particularly raised by R2, which need to be addressed. Particularly, I agree with R2 about the use of the term "accuracy" which seems to change its meaning across sections. Please be consistent!

Thank you again. I believe after these minor changes, we can move forward to production.

·

Basic reporting

No comment

Experimental design

No comment

Validity of the findings

No comment

Additional comments

In the initial round of reviews, I raised a number of concerns and recommendations and these have all been satisfactorily addressed in the revised manuscript.

Reviewer 2 ·

Basic reporting

no comment

Experimental design

no comment

Validity of the findings

no comment

Additional comments

Please see the attached pdf

Annotated reviews are not available for download in order to protect the identity of reviewers who chose to remain anonymous.

---

## Round 0.3 · accepted · Accept

· Academic Editor

Accept

Both reviewers are satisfied with the work done by the authors.